# Effect of Different Milling Media for Surface Coating on the Copper Powder Using Two Kinds of Ball Mills with Discrete Element Method Simulation

**Amgalan Bor [1], Battsetseg Jargalsaikhan [1,2], Jehyun Lee [1,2] and Heekyu Choi [2,3,*]**

[1] Engineering Research Center (ERC) for Integrated Mechatronics Materials and Components, Changwon National University, Changwon, Gyoungnam 641-773, Korea; amgalanbor@gmail.com (A.B.); battsetseg12@yahoo.com (B.J.); ljh@changwon.ac.kr (J.L.)

[2] Graduate School of Material Science Engineering, Changwon National University, Changwon, Gyoungnam 641-773, Korea

[3] Department of Mechanics Convergence, College of Engineering, Changwon National University, Changwon, Gyoungnam 641-773, Korea

*   Correspondence: hkchoi99@changwon.ac.kr; Tel.: +82-55-213-3841

**Abstract:** This study investigated the effect of three different ball materials on the metal-based carbon nanotube (CNT) composites used as surface coatings on metal-powder to fabricate high-quality nanocomposites. The effect of ball material, different rotation speeds, and milling times on the coating characteristics of the metal-based nanocomposite were studied. The mechanical dry coating was used to fabricate CNT coatings on the surface of copper powder particles via two different ball milling machines such as a traditional ball mill and a stirred ball mill. We explored the effect of the milling media of the ball mill under different ball materials and ball sizes on the metal powder during the ball milling process with DEM simulation. Using discrete element method simulation to obtain the average velocity, force, and, kinetic energy of the milling media in a low and high energy ball mills.

**Keywords:** dry surface coating; ball mill; copper; carbon nanotube; milling media; DEM simulation

---

## 1. Introduction

In recent years, nanocomposites have been fabricated by means of the mechanical alloying (MA) process that has been developed for obtaining metal matrix nanocomposites through the dry surface coating of metal powders [1–3]. MA is very simple process and generic term for processing of metal powders in high energy ball mills and economically viable process with important technical advantages. The greatest advantages of MA is carried out at room and near temperature and these applications is in the production of a highly homogeneous product without any segregation effects and synthesis of novel alloys etc. [4,5].

The particle coating is used to modify the properties of powders by using mechanical forces to attach the guest particles to the surface of the host particles. The coating of fine guest particles onto the surface of host particles is accomplished through shear and impact forces generated during the ball milling process [2,6,7]. The use of ball milling machines for the fabrication of metal-based nanocomposites has been extensively investigated [8–11]. In the ball milling process, the milling balls collide repeatedly with the powder mixture placed in the ball mill. The milling process can be performed using various types of ball mills, namely, planetary, traditional, horizontal, and attritor ball mills [12].

The ball milling of powders, a quite mature mechanical processing method, has recently been used for new applications, e.g., mechanical coating of surfaces [13,14]. Owing to their fascinating

properties, composite materials based on metal matrix have received a great deal of attraction found to be suitable for various applications. Cu/CNT nanocomposites have a highest thermal and electrical conductivity properties, it has many kinds of application such as electrical, automotive, electronics and telecommunication industry, high performance heat sink materials, and household products [15–17].

Discrete element method (DEM) is one of the most popular techniques for simulating and analyzing the solid particle behavior [18]. DEM is a numerical method for computing the motion and collisions of particles. DEM determine Newton's equations of motion to resolve particle motion and using a contact law to resolve inter-particle contact forces [19].

In our previous work, we studied the dry ball milling method to CNT coat on the metal surface with zirconia milling media using two kinds of ball milling machines, it can be seen good coating properties on the low rotation speed and long milling time [2]. From this research, we obtained the fabricating Cu/CNTs nanocomposite with various experimental conditions, results show that the dependence on the ball milling mechanism. This prompt us to investigate the effect of three different ball materials such as zirconia, aluminum and stainless-steel ball on the coating properties. This article determined the effect of three different milling media such as zirconia, alumina, and stainless steel on the mechanical coating process associated with each milling machine.

## 2. Materials and Methods

The copper powders with a purity of 99.0%, median particle size of 25 μm and multi-walled carbon nanotubes with 5 μm in average length and 20 nm in diameter used in this study shown in Figure 1. The mechanical dry coating process was performed using a traditional ball mill (TBM) and a stirred ball mill (SBM) with a stainless-steel pot and a Teflon (PTFE) coated pot, respectively. SBM is vertical type and high-speed ball mills, are the mills in which large quantities of powder. TBM type is horizontal rotating, continuous operation, and low-energy consumption [2,10]. The mechanical coating was carried out at different revolution speeds such as 50, 100, and 300 rpm for milling times of 12 or 48 h in each mill [20] (Table 1). Milling media was used three kinds of 5 mm of milling media, such as alumina, zirconia, and stainless-steel balls were employed in each case. Figure 2 shows the schematic picture of the CNT coating on the metal surface powders by two kinds of ball mill milling process in this study. The results were analyzed via scanning electron microscopy (SEM) and field emission scanning electron microscopy (FESEM), respectively. The three-dimensional motion of the balls during the two kinds of ball milling process by the DEM simulation (Table 2).

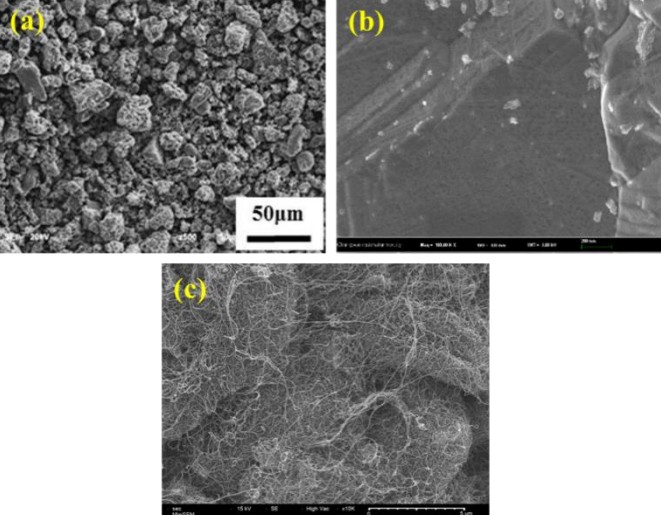

**Figure 1.** (**a**) SEM micrographs of the raw material copper (Cu), (**b**) FESEM micrographs showing the surface of the Cu powder and (**c**) multi-walled carbon nanotube (MWCNT).

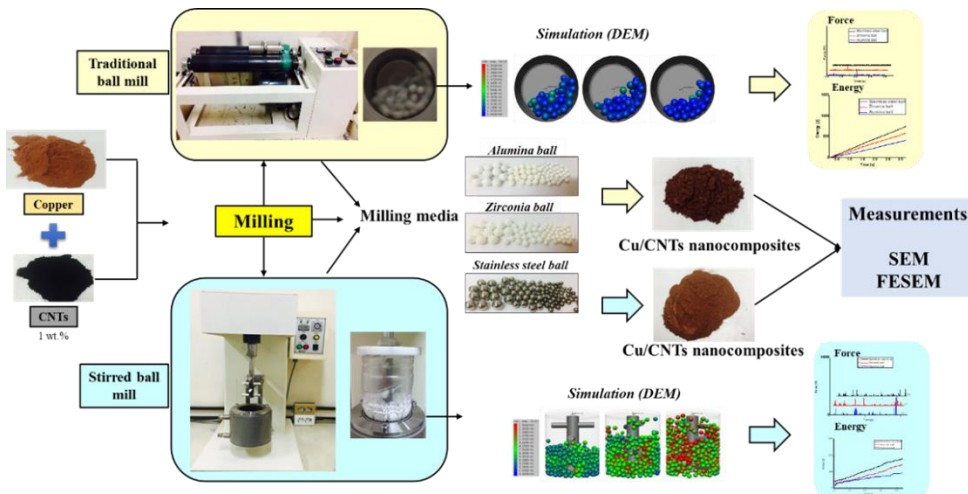

**Figure 2.** Schematic view of the CNT coating on the metal surface powders by two kinds of ball milling process [20].

**Table 1.** Experimental conditions for CNT coating on the metal surface by ball milling processes.

| Coating Experimental Conditions | TBM/SBM |
|---|---|
| Revolution speed (rpm) | 50, 100 and, 300 |
| Ball size (mm) | 5 |
| Milling time (h) | 12/48 |
| Ball powder ratio | 10:1 (fixed) |
| Ball filling ratio [-] | 0.3 (fixed) |
| Amount of CNTs [wt%] | 1 |
| Ball material | Alumina, Zirconia, and Stainless-steel ball |
| Temperature | Room temperature |

**Table 2.** Simulation conditions of the TBM and the SBM.

| Simulation Conditions | | TBM | | SBM |
|---|---|---|---|---|
| | | Ball to Ball | Ball to Wall | |
| Friction coefficient (-) | Alumina media | 2.0 | 0.1 | 0.9 |
| | Zirconia media | 1.0 | 0.5 | 0.8 |
| | Stainless steel media | 0.3 | 0.1 | 0.7 |
| Revolution speed (rpm) | | 50, 100, 300 | | |
| Ball size (mm) | | 5 | | |
| Number of the media | 5 mm | 99 | | 1490 |
| Density of media [g/cm$^3$] | alumina | 4.36 | | |
| | zirconia | 6.22 | | |
| | stainless steel | 7.95 | | |
| Ball filling ratio [-] | | 0.3 | | |
| Young modulus [MPa] | | 200,000 | | |
| Poisson's ratio | | 0.3 | | |
| Thermal conductivity [W/m K] | | 3.5 | | |

## 3. Results and Discussion

The SEM results show the particle size and shape of copper based CNT nanocomposites ball milled with three kinds of milling media such as alumina balls, zirconia balls, stainless steel balls and under the various experimental conditions using two different ball milling machines. Figure 3a,b

show the effect of the ball materials on the shape of Cu/CNT nanocomposites for both mills (TBM and SBM). The particle shape remained unchanged during 12 h of milling using TBM and SBM. During prolonged milling (48 h), the SBM with three different milling media yielded plate-type shape (Figure 3d). Figure 4a,b shows that 12 h of milling with the two different ball milling machines yielded the same shape. However, the particle shape of the SBM-processed samples changed to plate-type when the milling time was increased (Figure 4d). Figure 4 shows that the particle shape changed from massive to plate-type with increasing revolution speed and milling time (100 rpm and 48 h), for the SBM. Figure 4b–d show that the particle size of the SBM-processed nanocomposite increased with increasing milling time and the particles became plate-like. Two different ball mills, a TBM and a SBM, were used in this study. These mills differ in their capacity, speed of operation, vertical and horizontal rotation, ball movement, and milling energy.

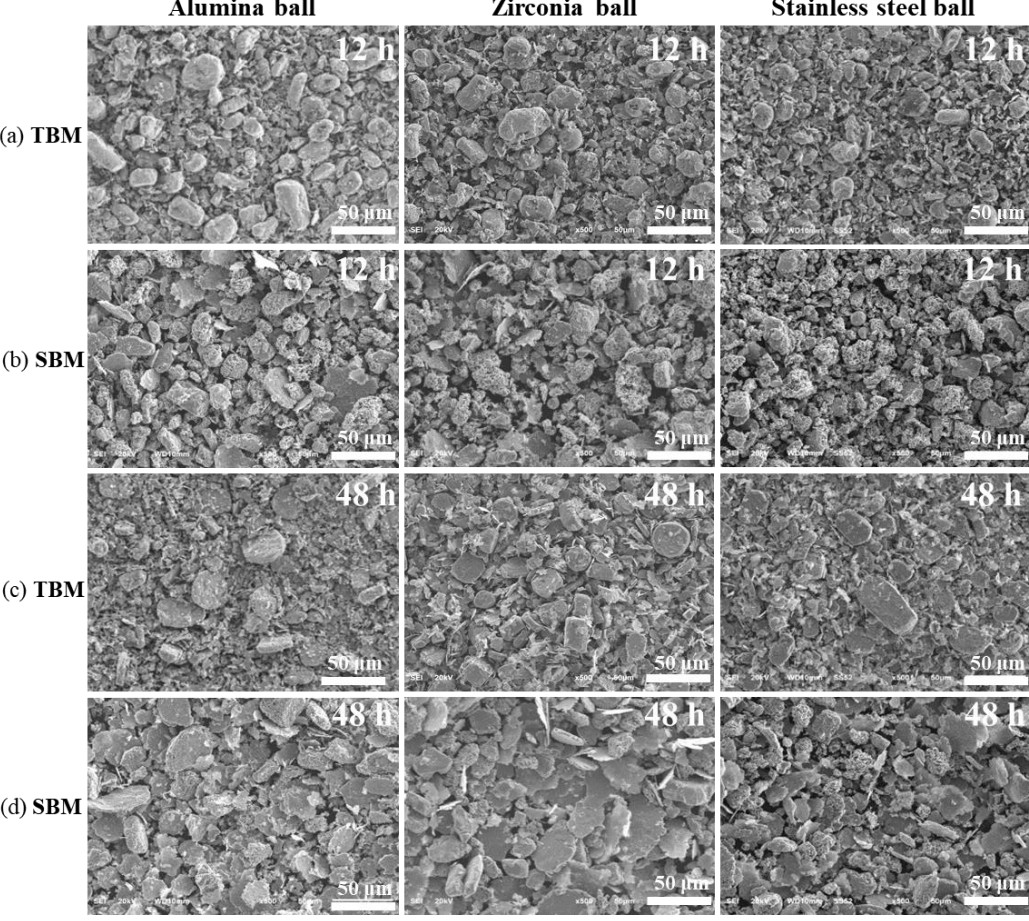

**Figure 3.** SEM images showing morphologies of Cu/CNT nanocomposites during exposure to different milling media (**a**) 12 h TBM, (**b**) 12 h SBM, (**c**) 48 h TBM, and (**d**) 48 h SBM at 50 rpm.

| Alumina ball | Zirconia ball | Stainless steel ball |
|---|---|---|

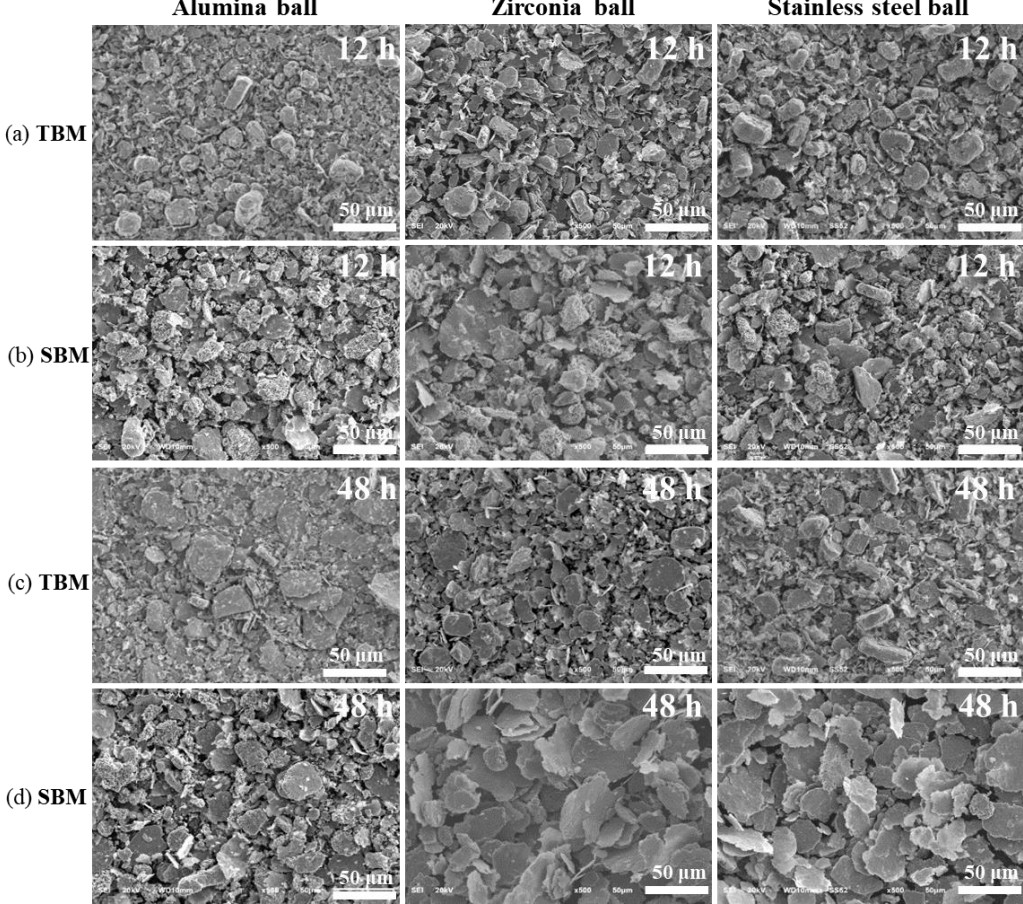

**Figure 4.** SEM images showing morphologies of Cu/CNT nanocomposites during exposure to different milling media (**a**) 12 h TBM, (**b**) 12 h SBM, (**c**) 48 h TBM, and (**d**) 48 h SBM at 100 rpm.

Furthermore, a SBM can also handle greater quantities of powder than planetary mills or TBM. Traditional ball mills, which rotate horizontally, are characterized by continuous operation and low-energy consumption. The ball behavior of TBM is significantly different from that of SBM, which transfers much greater energy, resulting in faster changes in particle size and shape [2]. As Figure 5 shows the particles changed to plate-like when the speed of revolution was increased to 300 rpm in both the TBM and SBM. The SEM images shown in Figure 5b–d revealed that the Cu/CNT nanocomposite particles are flattened (plate type) in the early steps of mechanical alloying (MA). In before study, we studied the raw-powder properties of CNT surface coatings and evaluated the effect of these properties on Cu powder using two different ball mills, a TBM and a SBM. The results revealed that the particle shape became plate-like with increasing revolution speed [2]. Figure 5 shows particle shape changes and demonstrates that unlike pure Cu powders, Cu/CNT nanacomposite was a slight size reduction rather than agglomeration. The results obtained for the mixture powders indicate that the CNTs used in this work acted as grinding aids to prevent particle agglomeration. Results different from those studied by Wang, Choi et al., those authors found that mechanical alloying is accompanied by marked agglomeration of CNT-Al composite [9].

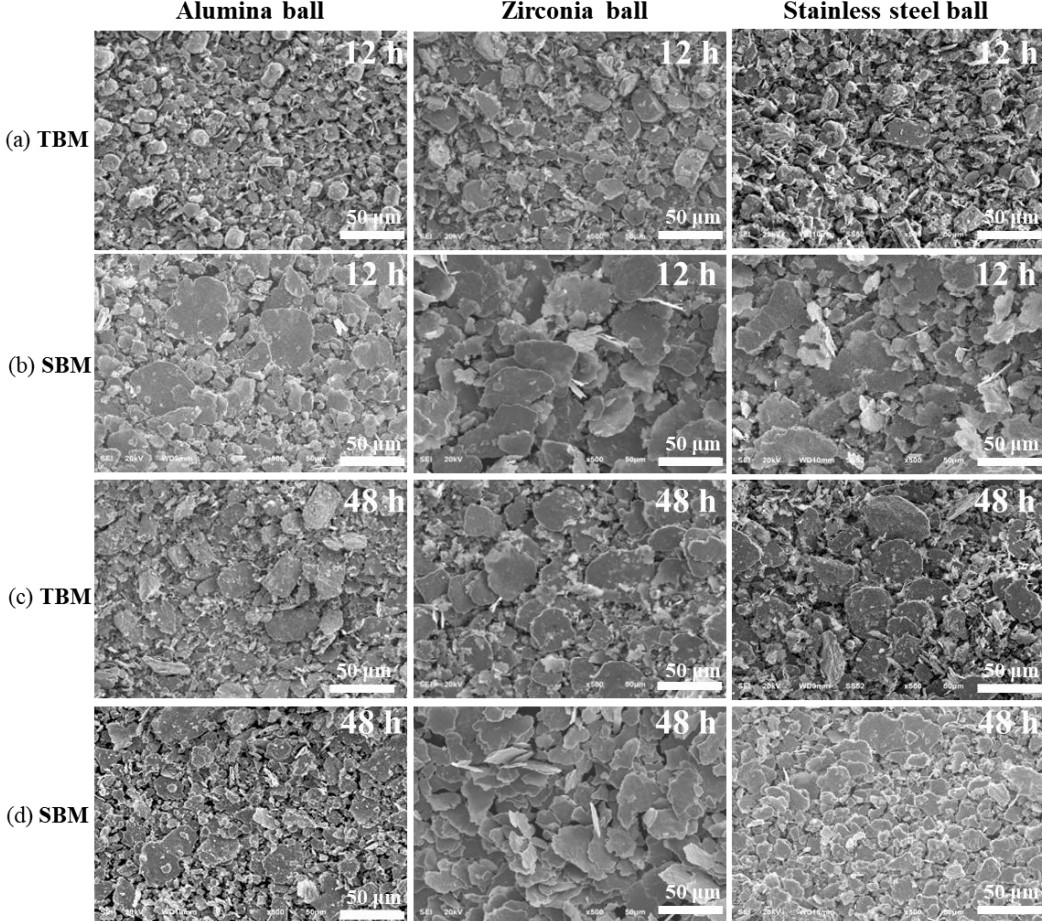

**Figure 5.** SEM images showing morphologies of Cu/CNT nanocomposites during exposure to different milling media (**a**) 12 h TBM, (**b**) 12 h SBM, (**c**) 48 h TBM, and (**d**) 48 h SBM at 300 rpm.

The FESEM micrographs in Figures 6–11 show the microstructures obtained when the Cu/CNT nanocomposite is milled under various conditions such as revolution speed, milling time, milling media (alumina, zirconia, stainless steel balls) in two different ball mills.

Figure 6 shows the nanocomposites ball milled after 12 h at a low revolution speed of 50 rpm. The CNT coated on the Cu powder particles obtained via TBM and three different milling media are shown in Figure 6a. Figure 6b show that the CNTs were well coated (amount of CNT) on the surface of the Cu particle when the using SBM. After 48 h of milling, the CNTs were spread on the surface of copper powder and coated at 50 rpm as shown in Figure 7. The CNT coating obtained after milling of 48 h in the TBM was more coated at 50 rpm when using three different milling media (see Figure 7a). Compared with the TBM, the SBM yielded a significantly lower number of CNTs, as shown in Figure 7b.

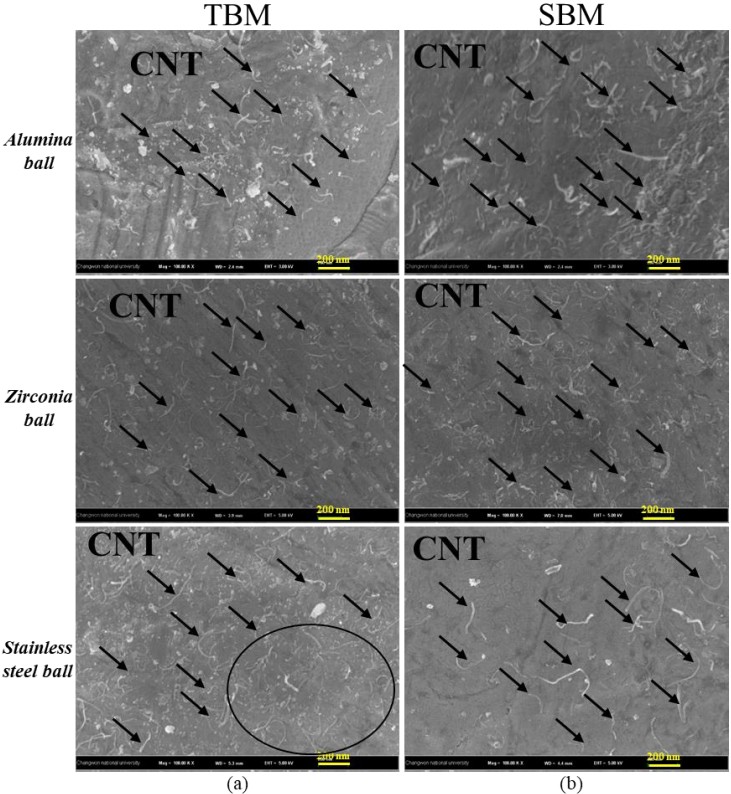

**Figure 6.** FESEM images (100,000×) of Cu/CNT nanocomposites for 12 h at 50 rpm in both mills (**a**) TBM and (**b**) SBM with three different milling media.

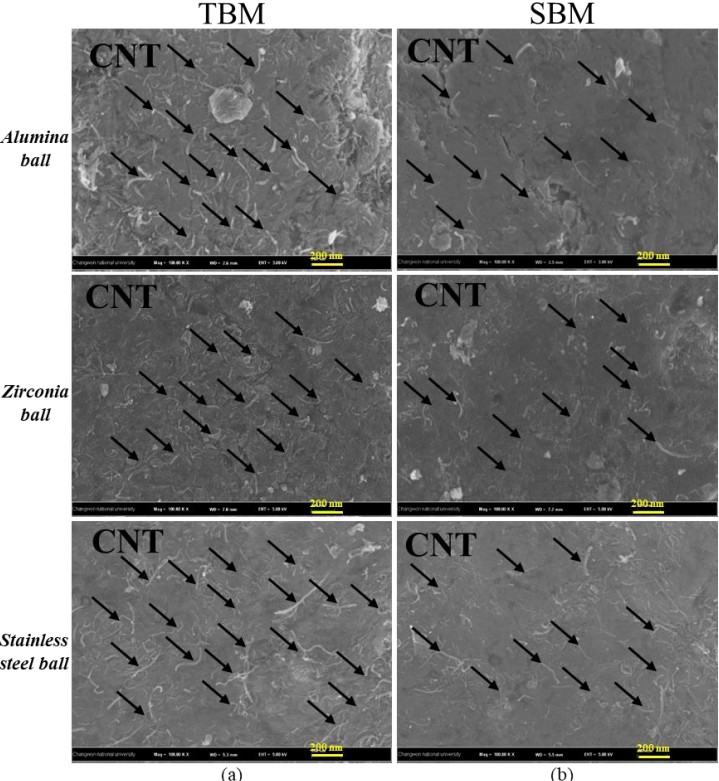

**Figure 7.** FESEM images (100,000×) of Cu/CNT nanocomposites for 48 h at 50 rpm in both mills (**a**) TBM and (**b**) SBM with three different milling media.

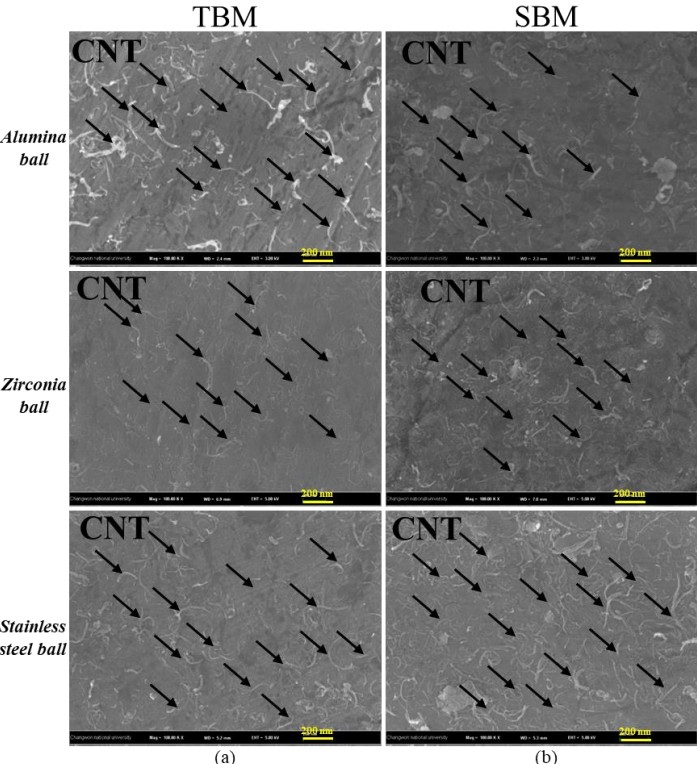

**Figure 8.** FESEM images (100,000×) of Cu/CNT nanocomposites for 12 h at 100 rpm in both mills (**a**) TBM and (**b**) SBM with three different milling media.

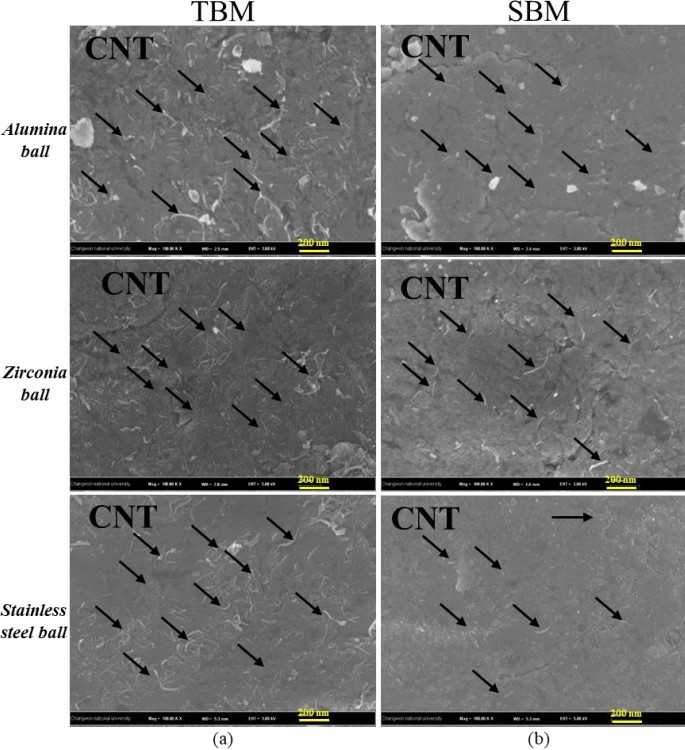

**Figure 9.** FESEM images (100,000×) of Cu/CNT nanocomposites for 48 h at 100 rpm in both mills (**a**) TBM and (**b**) SBM with three different milling media.

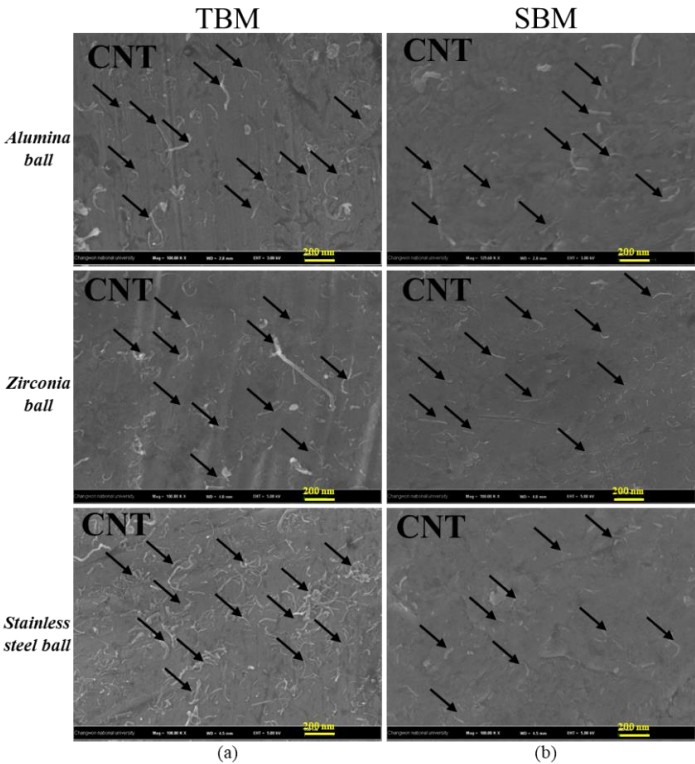

**Figure 10.** FESEM images (100,000×) of Cu/CNT nanocomposites for 12 h at 300 rpm in both mills (**a**) TBM and (**b**) SBM with three different milling media.

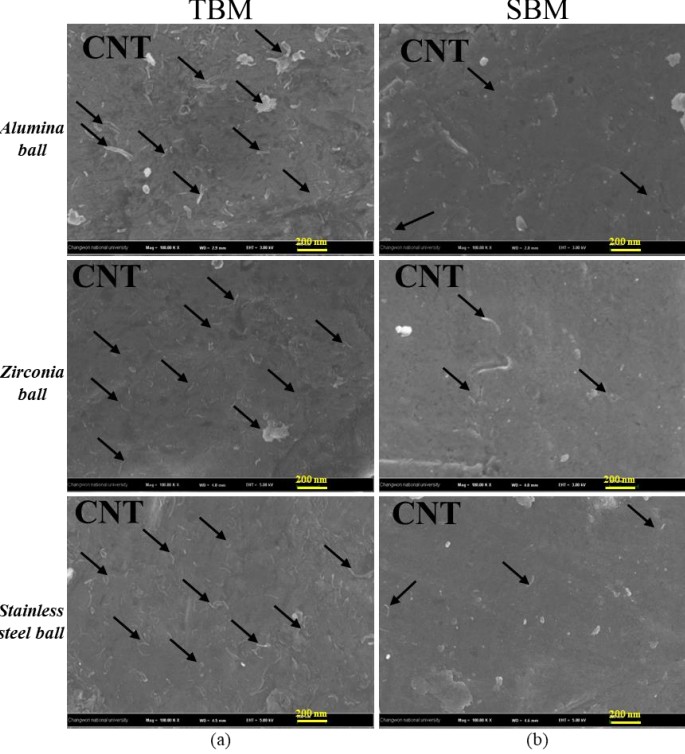

**Figure 11.** FESEM images (100,000×) of Cu/CNT nanocomposites for 48 h at 300 rpm in both mills (**a**) TBM and (**b**) SBM with three different milling media.

Figures 8 and 9 show the Cu/CNT nanocomposites milled for 12 h and 48 h at revolution speed of 100 rpm in both mills. In case of SBM, the CNT well coated on the Cu particles obtained after milling 48 h at 100 rpm as shown in Figure 8b. The CNTs were coated on the Cu particles after 12 h milling in TBM in Figure 9a. Figure 10 shows that, for 12 h of milling at a high revolution speed of 300 rpm, the TBM yielded a better CNT coating than the SBM. At this revolution speed, the CNTs coated the surface of the Cu powder and became embedded within the surface after 48 h milling period, as shown in Figure 11. Due to the high contact number between the ball and the particles in TBM, long milling at low revolution speeds produces well-coated Cu particles [2]. For SBMs with more impact than occur in TBM, a uniform CNT coating was obtained after 12 h at low revolution speed. In addition, the CNT coating obtained at low revolution speed (50 rpm) was greater to the high revolution speeds such as 100 rpm and 300 rpm. This is generally due to the cracks created at low revolution speeds on the surface of the Cu particles. Cu powder and CNTs were mechanically alloyed by dry ball milling process, which is based on van der Waals interactions between the host particles (Cu) and guest materials (CNTs) [20]. This coating experiment was carried out at revolution speeds of 50, 100 and 300 rpm and the successful CNT coating on the metal powder particle at 50 rpm in a TBM and SBM.

The important factor in CNT coating on the metal matrix is subjected to a small impact force of milling balls over a longer time, rather than a large force concluded a shorter time. The amount of contact during the ball milling plays a greater role than the impact power associated with the milling ball.

Longer milling at low revolution speed yielded well coated copper particles that were better coated than those obtained after shorter milling, owing to the higher contact number between the milling balls and the particles in the TBM. SBM has a stronger impact power compared to TBM, uniform CNT coatings were obtained after short milling of 12 h at a low revolution speed. A better CNT coating on the metal-powder surface was obtained in the TBM (than in the SBM), because the balls moved inward from the edge to the center of the pot during the milling process [20].

Figure 12 shows a schematic of the mechanism associated with CNT coating of a metal powder. The coating mechanism consists of three different processes, where CNTs spread over the surface of the copper particles, then become partially embedded in the Cu matrix, and subsequently become completely embedded in the matrix [20]. In this study, we investigated the effect of three milling media on the CNT coating of a metal surface. For a low revolution speed (50 rpm), only slight differences were observed when three different media materials (alumina, zirconia, and stainless steel) were employed for milling times of 12 and 48 h in both mills. The metal surface was well coated with CNT during 12 h of milling when the stainless steel balls and alumina balls were employed at a revolution speed of 100 rpm (Figure 8a,b). However, each ball material remained virtually unchanged during milling. This resulted from the fact that copper (Cu) powder is a ductile material that induces changes in the coating properties. In our previous work, we investigated the effect of milling media on the shape of Cu powders milled for various times at different revolution speeds. Three types of milling media with differing sizes and component materials were considered, and the ball behavior was investigated via DEM simulations [21]. The results revealed that, during the milling process, the size of the milling media changed, whereas the material composing the media remained unchanged.

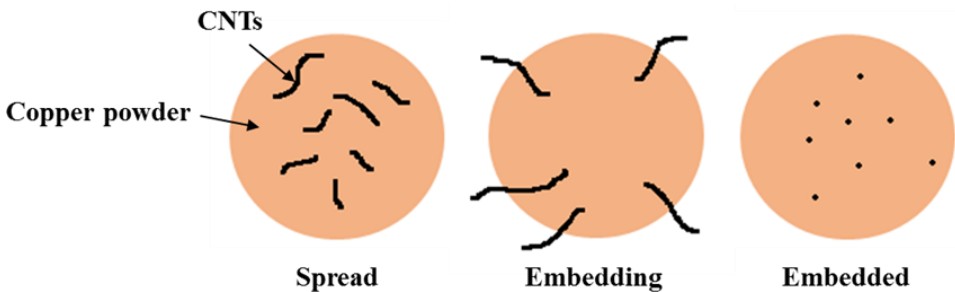

**Figure 12.** The schematic picture of mechanism of the CNT coating on the metal surface [20].

Milling media are simulated as discrete elements and the compression and shear forces for each impact are determined by Hertz and Mindlin's contact model, and the resulting velocity is calculated according to Newton's law of motion in the DEM simulation [22]. In order to explore the motion of the balls in a ball milling process, we performed the ball impact energy estimated by a DEM simulation. The DEM simulations can be obtained quantitative results such as force, impact energy, torque, power, average velocity, collisions number, rising height etc. In our case, we can obtained force, energy, torque, power, average velocity, and ball motion of the ball from DEM simulation.

Table 3 show the total impact force of balls obtained by DEM simulation results in both mills such as TBM and SBM. It can be seen big difference between results of TBM and SBM. SBM is high energy ball milling machine and there has a big pot compared to other ball milling machines. In the DEM simulation, number of the media are 99 and 1490 in both mills, respectively. The mean particle velocity increases monotonically with milling media due to increased contacted between balls with increasing number of balls [23]. The effect of milling media on the simulation result was analyzed quantitatively by DEM simulation, it was not noticeably changed in different milling media such as alumina, zirconia, and stainless steel. Revolution speed is increases, milling media have higher velocities, which results in more frequent and stronger collisions in a DEM simulation [23].

**Table 3.** Total impact force obtained by DEM simulation in both mills TBM and SBM.

| Revolution Speed (rpm) | Ball Type | TBM | SBM |
|---|---|---|---|
| | | Total Force (N) | |
| 50 | Alumina ball | 250 | 15,534 |
| | Zirconia ball | 225 | 13,292 |
| | Stainless steel ball | 280 | 16,758 |
| 100 | Alumina ball | 265 | 28,712 |
| | Zirconia ball | 277 | 29,451 |
| | Stainless steel ball | 339 | 33,402 |
| 300 | Alumina ball | 301 | 64,597 |
| | Zirconia ball | 307 | 65,538 |
| | Stainless steel ball | 338 | 68,088 |

Figures 13 and 14 show the impact energy of the balls in two kinds of ball mills such as TBM and SBM has been calculated by the DEM simulation of balls motion, respectively. It can be seen that the picture of real ball movement and a snapshot of the simulation using three different milling media with the revolution speed of 50, 100, and 300 rpm at each ball mill. Snapshot pictures show that the similar between real experimental movement and simulated movements. Figures 15 and 16 show the quantitative results of forces and impact energies for three kinds of milling media with various experimental conditions. The results suggest that both total force and impact energy increase as the revolution speed increases. In our case, it was not found the contact number of the balls, we determined only force and energy of the balls during the ball milling process. DEM simulation obtained the collision frequency of the balls, the kinetic energy of the balls the total force between balls, the motion and other information about the movements of the balls inside ball mill.

Ball mills use balls of various specific gravity (such as ceramic, metal) as milling media and transfer the energy of the ball into the sample in different ways, such as impact, shear, and friction. There are various kinds of balls used for milling media (cast iron, forged steel, cast steel, steel flint, ceramics, tungsten carbide ball) [24]. Comparing the effects of different kinds of milling media on the total impact energy, it was obtained slight differences from the DEM simulation. Thus, the results of this study were similar to those of previous study [25]. There is a well-defined correlation between the impact energy and the revolution speed on the material of the milling media, and it has been found the milling speed increases as the impact energy increases.

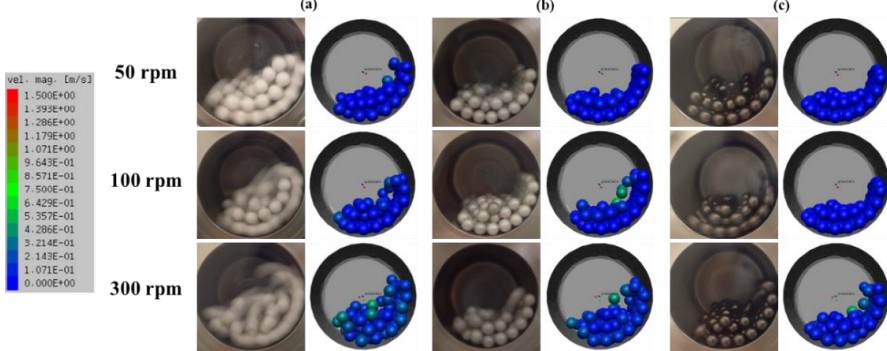

**Figure 13.** Simulation and actual snapshot photograph of the media motion by DEM simulation (**a**) alumina ball, (**b**) zirconia ball, (**c**) stainless steel ball with different revolution speeds in a TBM.

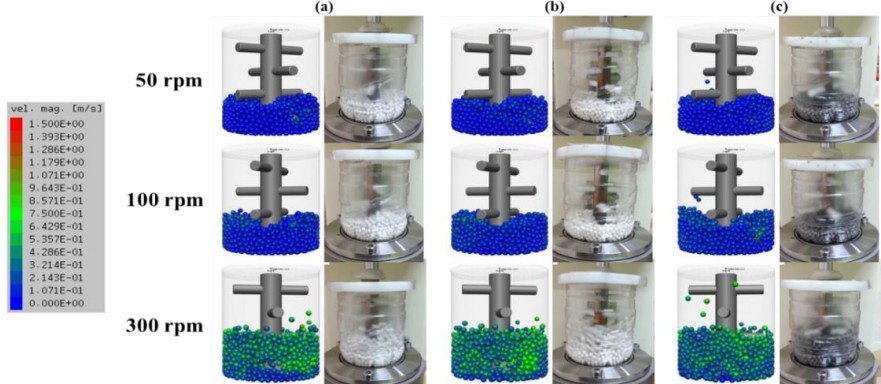

**Figure 14.** Simulation and actual snapshot photograph of the media motion by DEM simulation (**a**) alumina ball, (**b**) zirconia ball, (**c**) stainless steel ball with different revolution speeds in a SBM.

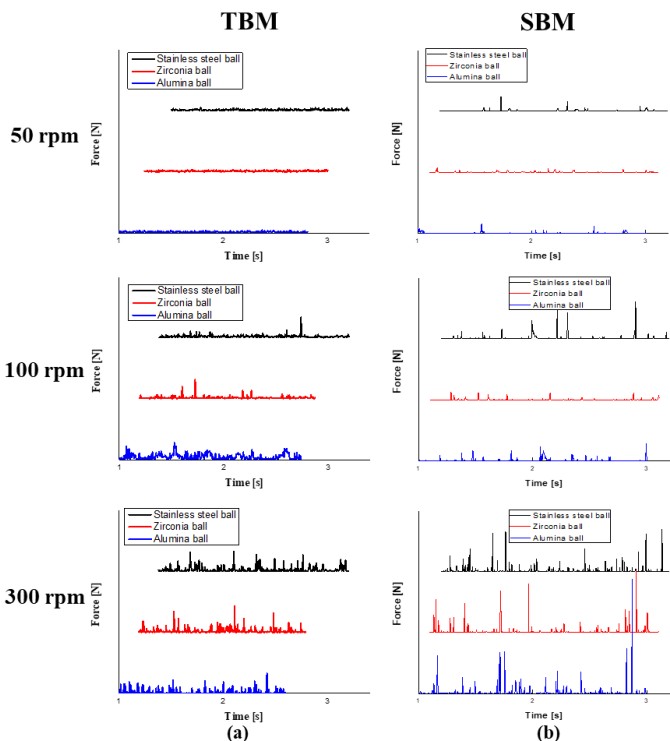

**Figure 15.** Milling force spectra of 50, 100, 300 rpm in both mills (**a**) TBM, and (**b**) SBM by DEM simulation.

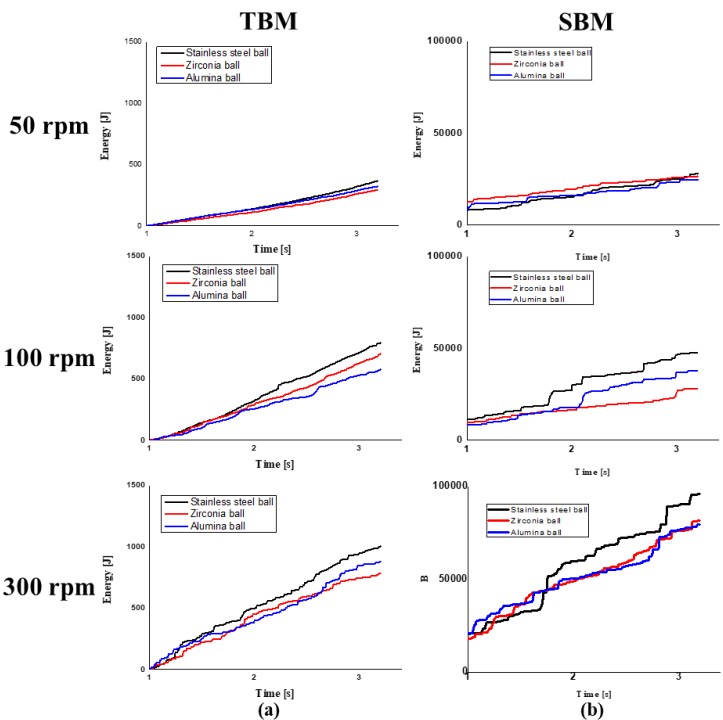

**Figure 16.** Milling energy spectra of 50, 100, 300 rpm in both mills (**a**) TBM, and (**b**) SBM by DEM simulation.

## 4. Conclusions

We investigated the primary-particle properties of CNT coatings and then, via two kinds of ball milling machines such as TBM and SBM, evaluated the effect of properties for the metal surface of the copper. The particle sizes and the particle morphologies of the Cu/CNT nanocomposites increased and changed from massive type to plate-like type with increasing revolution speed and milling time, respectively. The FESEM results showed that a low revolution speed (50 rpm) and a long milling time (48 h) provide the optimal conditions for obtaining Cu/CNT nanocomposites in the TBM. In the case of the SBM, the best coatings were obtained after 12 h at 50 rpm. In the coating experiment, the coating property and the amount of contact exerted more influence than the impact forces, owing to the coating on the metal surface of the Cu powder. The TBM yielded better CNT coatings than the SBM. This resulted from the fact that, in the TBM, the balls moved inward from the edge to the center of the pot during the milling process. Generally, each ball material changed only slightly during the experiment, because Cu powder is a ductile material that changes the coating properties. From the DEM simulation, it was obtained the force and energy of balls and ball behavior in the ball mill. Impact forces have less influential in the coating property and the contact number is more influential in coating experiment. The ball material properties affect no more difference in the coating.

**Author Contributions:** Conceptualization and experiment guidance, H.C.; manuscript writing, data analysis, A.B.; carried out experiments; data collection, A.B.; study design, data analysis, A.B. and B.J.; project administration, J.L.; funding acquisition, J.L. All authors have read and agreed to the published version of the manuscript.

**Funding:** This work was supported by the National Research Foundation of Korea (NRF) with a grant funded by the Korean government (MSIP) (2018R1A5A6075959).

**Conflicts of Interest:** The authors declare no conflict of interest.

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
