# Peer review of "Effect of Different Milling Media for Surface Coating on the Copper Powder Using Two Kinds of Ball Mills with Discrete Element Method Simulation"

_coatings, doi:10.3390/coatings10090898_

Round 1

Reviewer 1 Report

The level of English in the manuscript needs to be improved, because in places it was difficult to understand the discussion presented by the authors.  In addition, a number of the conclusions drawn in the manuscript do not seem to be clearly supported by the micrographs presented.  This needs to be addressed before acceptance of the paper for publication.  Furthermore, the red scale markings on the micrographs are hard to read.  A different colour would aid legibility.

More specific comments are as follows:

Line 92.  It is not clearly evident that plate-type morphology is produced by the TBM at 48 h and 100 rpm n Figure 4.

Lines 110-112.  It looks like plate-type morphology is only produced at 48 h for the TBM at 300 rpm in Figure 5.

Lines 118-119.  Data is not provided to support the statement that the best CNT coatings are obtained after a ling milling time (48 h) at a low rotation speed (50 rpm).

Lines 120-121.  The statement that plate-like morphologies were obtained when the milling time was increased to 48 h does not appear to be correct.  The SBM also produced plate-like morphologies after 12 h.

Lines 135-137.  This needs further explanation and what constitutes "well coated"..

Line 138.  What is the definition of homogeneous coating on the surface at 50 rpm?  This does not appear to be the case from the micrographs.

Lines 138-140.  This sentence does not make sense.  Please rewrite.

Lines 149-150.  This statement is not at all obvious.  Please explain.

Lines 156-158.  Please explain why the Cu/CNT nanocomposite particles are superior.

Lines 161-163.  Coatings were also evident at high rotation speed.

Lines 183-185.  A definition of better coating would aid comprehension.  It is not immediately obvious from the micrographs.

Lines 195-197.  Why select stainless steel rather than say alumina?

DEM Simulations.  The simulations do not appear to improve the understanding of coating performance. 

These comments need to be addressed prior to deciding on publication of the paper.

Author Response

Response to Reviewer 1 Comments

The level of English in the manuscript needs to be improved, because in places it was difficult to understand the discussion presented by the authors. In addition, a number of the conclusions drawn in the manuscript do not seem to be clearly supported by the micrographs presented. This needs to be addressed before acceptance of the paper for publication. Furthermore, the red scale markings on the micrographs are hard to read. A different colour would aid legibility.

First of all, thank you very much for your suggestion. English editing service already has done by Asia Science Editing for this paper. We have checked the grammar of English more again. We have reorganized the SEM images and corrected red scale markings on the micrographs.

Point 1: Line 92. It is not clearly evident that plate-type morphology is produced by the TBM at 48 h and 100 rpm n Figure 4.

Response 1: We have corrected the explanation in line 92. Fig. 4(c) is not clearly plate-type morphology compared to Fig. 4(d).

Point 2: Lines 110-112. It looks like plate-type morphology is only produced at 48 h for the TBM at 300 rpm in Figure 5.

Response 2: If you look closely, you can see that the particle morphology has changed to a plate-type morphology in Figure 5(b-d). 

Point 3: Lines 118-119.  Data is not provided to support the statement that the best CNT coatings are obtained after a long milling time (48 h) at a low rotation speed (50 rpm).

Response 3: I agree with your suggestion. We have corrected this sentence in lines 118-119.

Point 4: Lines 120-121. The statement that plate-like morphologies were obtained when the milling time was increased to 48 h does not appear to be correct. The SBM also produced plate-like morphologies after 12 h.

Response 4: We revised this sentence in lines 120-121.

Point 5: Lines 135-137. This needs further explanation and what constitutes "well coated".

Response 5: We add more explanation about well coated in line 135-137. Compared with figure 6(a), there is a lot of CNT on the copper particle. Well coated means amount of CNT.

Point 6: Line 138. What is the definition of homogeneous coating on the surface at 50 rpm? This does not appear to be the case from the micrographs.

Response 6: We deleted homogeneous from this sentence in line 138.

Point 7: Lines 138-140. This sentence does not make sense. Please rewrite.

Response 7: We revised this sentence in lines 138-140.

Point 8: Lines 149-150. This statement is not at all obvious. Please explain.

Response 8: We revised this sentence in lines 149-150.

Point 9: Lines 156-158. Please explain why the Cu/CNT nanocomposite particles are superior.

Response 9: We revised this sentence in lines 156-158.

Point 10: Lines 161-163. Coatings were also evident at high rotation speed.

Response 10: This paper investigated the effect of the raw powder properties of carbon nanotube surface coating on the Cu particles with three different milling media using a ball milling process. Results show the CNT coating on the metal powder particles, it can be seen well-coated results obtained at low rotation speed compared to high rotation speed using a two different ball milling machines. But there have two different milling machines, it was obtained at a long time (48 h) and a short time (12 h) in a TBM and SBM, respectively.

We revised this sentence in lines 161-163.

Point 11: Lines 183-185. A definition of better coating would aid comprehension. It is not immediately obvious from the micrographs.

Response 11: We have defined the optimal experimental conditions of CNT surface coating on the Cu metal particles for the traditional ball mill. The traditional ball mill can be used for a long time and also saves energy. 

Point 12: Lines 195-197. Why select stainless steel rather than say alumina?

Response 12: We revised this sentence in lines 195-197. 

Point 13: DEM Simulations. The simulations do not appear to improve the understanding of coating performance. 

Response 13: In our case, we did quantitative analysis in a ball milling process using a discrete element method simulation. In order to explore the motion of the balls during ball milling process, we obtained the ball impact force and energy by a DEM simulation. But we can not define the contact number of the ball. In this study, it can be seen that the contact number was more influenced than the impact energy of the ball. We added the reason why we do simulation in the results and discussion section.

Reviewer 2 Report

The authors investigated the primary-particle properties of CNT coatings and  using two different ball mills (TBM and SBM) to evaluate the effect of these properties on the metal surface of the copper powder. There are lots of SEM images shown in the results, but barely other characteristics on the resulted composites. Mechanical or other characterizations should be added with a possible statistics analysis. Besides, for the simulation results, is it possible to add a quantitative way to show the results?

Author Response

Response to Reviewer 2 Comments

Point 1: The authors investigated the primary-particle properties of CNT coatings and using two different ball mills (TBM and SBM) to evaluate the effect of these properties on the metal surface of the copper powder. There are lots of SEM images shown in the results, but barely other characteristics on the resulted composites. Mechanical or other characterizations should be added with a possible statistics analysis. Besides, for the simulation results, is it possible to add a quantitative way to show the results?

Response 1: Thank you very much for your recommend. We have reorganized the SEM images in this paper. We checked only physical property on the grinding process for metal powder coating. We should check the product of mechanical property in the future. We added more explanation about DEM simulation in the results and discussion section and added the total impact force by quantitatively results obtained by DEM simulation in table 3.

Reviewer 3 Report

The submitted manuscript is an interesting study of the effect of three different ball materials on the raw-powder 14 properties of metal-based carbon nanotube (CNT) composites used as surface coatings on metal-15 powder to fabricate high-quality nanocomposites. The manuscript is written in high scientific level, the results of the study are presented logically and in understandible form. The manuscript will be interesting for a wide range of readers and difinetely it will have a lot of citations. Thus, I recommend to accept the submitted manuscript.

Author Response

Response to Reviewer 3 Comments

Point 1: The submitted manuscript is an interesting study of the effect of three different ball materials on the raw-powder 14 properties of metal-based carbon nanotube (CNT) composites used as surface coatings on metal-15 powder to fabricate high-quality nanocomposites. The manuscript is written in high scientific level, the results of the study are presented logically and in understandible form. The manuscript will be interesting for a wide range of readers and difinetely it will have a lot of citations. Thus, I recommend to accept the submitted manuscript.

Response 1: Thank you very much for your kindly recommend.

Reviewer 4 Report

This study investigated the effect of three different ball materials in order to fabricate high-quality nanocomposite of metal-based carbon nanotube (CNT).

The paper could be published sfter some revisions. The introduction should be implemented by references to the appliation of Cu-CNTs nanocomposites and the advantage of this method with respect to other ones.

Figures should be clearly introduced.

Why the authors did not add some chemical-physical characterizations of powders before and after the experiments? it could be important to knoe if the process affect the chemical structure of CNTs or Cu according to their application.

Author Response

Response to Reviewer 4 Comments

This study investigated the effect of three different ball materials in order to fabricate high-quality nanocomposite of metal-based carbon nanotube (CNT). The paper could be published after some revisions.

We appreciate the reviewer's advice. We responded point by point in detail to the reviewer's comments.   

Point 1: The introduction should be implemented by references to the application of Cu-CNTs nanocomposites and the advantage of this method with respect to other ones.

Response 1: We added about of application of Cu-CNTs nanocomposites and the advantage of this method in the introduction section.

Point 2: Figures should be clearly introduced.

Response 2: Figures were revised.

Point 3: Why the authors did not add some chemical-physical characterizations of powders before and after the experiments? It could be important to know if the process affect the chemical structure of CNTs or Cu according to their application.

Response 3: The goals of this investigation are CNT-coating for different kinds of ball material and comparison on the mill of DEM simulation results for different kinds of grinding mills. We checked only physical property on the grinding process for metal powder coating. The process affects the chemical structure of the product is not considered in this study. In the future, we should check the chemical structure study on review comments.

Round 2

Reviewer 1 Report

Thanks for considering the comments provided.  The revised draft is an improvement over the earlier draft, although there is still scope for improving the English grammar.

Reviewer 2 Report

Accept